# Genome-Wide Identification and Functional Characterization of *CesA10* and *CesA11* Genes Involved in Cellulose Biosynthesis in Sugarcane

**DOI:** 10.3390/ijms262211046

**Published:** 2025-11-14

**Authors:** Yi Xu, Nameng Qi, Yi Han, Liying Cai, Xue Wang, Heyang Shang, Qing Zhang, Jisen Zhang

**Affiliations:** 1State Key Laboratory for Conservation and Utilization of Subtropical Agro-Bioresources, Guangxi University, Nanning 530004, China; xuyi903@foxmail.com (Y.X.); qnm8646@163.com (N.Q.); hanyi2470@dingtalk.com (Y.H.); cailiying1230@foxmail.com (L.C.); wxue0107@foxmail.com (X.W.); heyangshang@hotmail.com (H.S.); zhangqing970@126.com (Q.Z.); 2College of Life Science and Technology, Guangxi University, Nanning 530004, China; 3College of Agriculture, Guangxi University, Nanning 530004, China

**Keywords:** cellulose biosynthesis, gene expression, sugarcane stem strength, secondary cell wall

## Abstract

Cellulose is the primary component of plant cell walls, and its content is linked to the strength of plant stems. The cellulose synthase genes (*CesA*) are crucial for regulating cellulose biosynthesis. To examine the characteristics and functions of *CesA* genes in sugarcane, our study conducted a genome-wide analysis of *the Saccharum officinarum LA-Purple genome*. The results identified 10 *CesA* genes in the *S. officinarum* genome, which could be grouped into six categories. *SoCesA10*, *SoCesA11*, and *SoCesA12* are clustered within the same subclass as genes involved in secondary cell wall synthesis in rice and Arabidopsis. Further transcriptome analysis of stems at different stages and sections showed that *SoCesA10*, *SoCesA11*, and *SoCesA12* were highly expressed during mature stages. Among these, *SoCesA10* and *SoCesA11* showed differences in expression between species and organs. Their gene functions were also validated in rice, revealing that the expression of *SoCesA10* and *SoCesA11* was positively correlated with cellulose content. In summary, this study identified key cellulose biosynthesis genes, *SoCesA10* and *SoCesA11*, in sugarcane and preliminarily confirmed their functions in rice, providing a foundation for breeding sugarcane with improved lodging resistance.

## 1. Introduction

Sugarcane (*Saccharum* spp.) is a perennial herbaceous crop in the Poaceae family, *Saccharum* L. genus. It is extensively cultivated in tropical and subtropical regions [1]. As an important economic crop cultivated in more than 90 countries worldwide, sugarcane exhibits typical C_4_ crop characteristics such as a high light saturation point, a low CO_2_ compensation point, strong stress resistance, and a high yield [2]. It not only supplies 75% of the world’s sugar, making it an important sugar crop, but also serves as a raw material for producing over 40% of the world’s ethanol, making it an important economic and energy crop [3].

Stalk strength is one of the important agronomic traits in sugarcane, as it affects the lodging resistance of the crop [4]. Lodging directly leads to a decrease in sugarcane yield and quality, as well as an increase in the difficulty of mechanical harvesting [5], resulting in more than 50% of the total cost attributed to manual labor in sugarcane harvesting [6]. These factors severely limit the development of the sugar industry and are among the urgent issues that need to be addressed in sugarcane production. Stalk strength is influenced by multiple factors, apart from diseases and pests, the most important being the intrinsic characteristics of the plant stalk itself, including the composition of the cell wall [7]. Among higher plants, the content of cellulose [4] and lignin [8] affects the degree of thickening of the cell wall, thereby determining the mechanical strength of the stalk. Plants can enhance secondary cell wall thickness at stem nodes and improve stem strength by regulating cellulose and lignin biosynthesis [9].

Plant cell walls have a complex structural composition, including polysaccharides, glycoproteins, and lignin, with cellulose being the main component [10]. The biosynthesis of cellulose microfibrils is catalyzed by the cellulose synthase complex (CSC), which consists of cellulose synthase subunits (CesAs) and several CesA-related proteins [11]. Different *CesA* genes have specific functions in cellulose synthesis, and multiple *CesA* genes work together in the process of cellulose biosynthesis [12,13,14,15]. While the number of these genes varies, they are clearly divided into two groups with different expression patterns based on their impact on the biosynthesis of the primary cell wall (PCW) or the secondary cell wall (SCW) [16] Based on previous research, the function of *CesAs* often aligns with the phylogenetic relationships with other orthologs in higher plants [17]. In Arabidopsis, the CesA protein family consists of 10 members, among which *AtCesA1*, *AtCesA3*, and *AtCesA6* serve as core components of the primary wall cellulose synthase complex. *AtCesA4*, *AtCesA7*, and *AtCesA8* are involved in secondary wall synthesis [18], while the core functions of *AtCesA10* and its roles in tissues like stems remain undefined. The rice CesA family comprises 11 members, and mutations in *OsCesA4*, *OsCesA7*, and *OsCesA9* reduce stem mechanical strength [19]. *OsCesA10* and *OsCesA11* are structurally unique, containing only typical cellulose synthase domains. Whether *OsCesA10* and *OsCesA11* participate in secondary wall cellulose synthesis to affect stem strength remains unknown.

Cultivated sugarcanes are derived from the hybridization of *Saccharum officinarum* and *Saccharum spontaneum*, both with highly complex genetic backgrounds. Recent studies have revealed that functional differentiation of *CesA* genes across monocot species plays a crucial role in stem strength and biomass accumulation, providing valuable insights for molecular breeding of lodging-resistant crops. However, a systematic investigation of the *CesA* gene family members in *Saccharum officinarum* and their functional validation remains limited. The molecular regulatory network of cellulose synthesis in sugarcane has not yet been elucidated, which seriously hinders the genetic improvement of sugarcane. While the high sugar content and stalk strength of modern sugarcane cultivars derive from *S. officinarum* [3]. Therefore, this study aims to identify cellulose enzyme genes in *S. officinarum* to provide genetic resources for sugarcane genetic improvement.

## 2. Results

### 2.1. Genome-Wide Identification of CesAs Genes in S. officinarum

In this study, using the sorghum CesAs gene as a reference, we identified 10 CesA genes in S. officinarum, which were named SoCesA1, SoCesA2, SoCesA3, SoCesA5, SoCesA7, SoCesA8, SoCesA9, SoCesA10, SoCesA11, and SoCesA12 (Table 1). The number of amino acids in the SoCesA protein ranges from 937 to 1374, with a protein molecular weight (MW) ranging from 102,705.03 to 154,548.22 Da, and a theoretical isoelectric point (pI) ranging from 6.12 to 8.69. The physical and chemical properties of the S. officinarum CesA family members were similar, and all members were located on the plasma membrane, which may be related to the biological functions of the genes.

Phylogenetic analysis (Figure 1A) reveals that the CesA family of seven species forms six clades. By studying CesA family members in different species, we can deduce the functions of homologous genes in other species. *CesA* gene structures (Figure 1B) show that, except for the significant differences in structure between *CesA3* and *CesA10* among species, the structures of other members are highly similar. Furthermore, conservation analysis of protein motifs (Figure 1B, Appendix A) shows that, except for the inconsistency in motif numbers of *CesA10* between the *S. spontaneum* and *S. officinarum*, there are almost no differences in other members of sugarcane between these two species. The CesA protein conserved motifs exhibit relative conservation among species and within the family (Figure 1C). This suggests that functional differences between *CesA* genes might be regulated at the upstream level.

### 2.2. The Expression Patterns of CesA Genes at Different Developmental Stages in S. spontaneum and S. officinarum

Transcriptional profiling was conducted on different organs at different developmental stages of the *S. spontaneum* and *S. officinarum* (Figure 2A). The results showed that in these two species, CesA2 was either not expressed or expressed at a low level. *CesA10*, *CesA11*, and *CesA12* exhibited clear organ-specific expression of the two ancestral species. Stem organs showed higher expression than leaves. *CesA1*, *CesA7*, and *CesA9* had the highest expression levels in leaves during the seedling stage for the two ancestral species, and stems also showed high expression in the seedling stage in *S. officinarum*. *CesA10*, *CesA11*, and *CesA12* genes had much higher expression in stems compared to leaves, especially during the mature stage, and also showed differential expression between species.

*CesA10* and *CesA11* were selected for further functional analysis due to their more pronounced differences between the species and organs. Real-time quantitative PCR (Figure 2B) confirmed that *CesA10* and *CesA11* were both highly expressed in stems across all three stages, with much higher expression in *S. spontaneum* compared to *S. officinarum*. The expression patterns were consistent with the trends observed in the transcriptome analysis.

### 2.3. The Subcellular Localization of SoCesA10 and SoCesA11

The full lengths of *SoCesA10* and *SoCesA11* were cloned from *S. officinarum* LA-purple. *SoCesA10* and *SoCesA11* had 3246 bp and 2934 bp open reading frames (ORF), respectively. Our study further investigated the subcellular localization of *SoCesA10* and *SoCesA11* in tobacco through infiltration transformation, using FM4-64 red fluorescence signal as a cell membrane marker. We observe GFP signals at the periphery of the cells, indicating that GFP fusions can be located at the cell wall, plasma membrane, or cytoplasm (Figure 3) and that, considering that previous CesA proteins from other species have been located at the plasma membrane [20], this suggests that sugarcane CesA proteins could display similar localization.

### 2.4. The Correlation Between the Expression Level of SoCesA and Cellulose Content

Through comprehensive transcriptome analysis and RT-qPCR validation, it was found that *CesA* family members in different organs and developmental stages are generally highly expressed in *S. spontaneum*, with higher expression levels in stems than in leaves. Cellulose content measurement results showed that there was no significant difference in cellulose content between the two original species in the seedling stage, but in the early and mature stages, the cellulose content in *S. spontaneum* was much higher than in *S. officinarum*, particularly prominent in the stems, consistent with the expression patterns in different organs (Appendix A).

During the early stages of maturity, there is no notable distinction in cellulose content among leaf species, but there is a noticeable difference in the stems, displaying a downward increasing trend. At the mature stage, regardless of the species, there is an overall reduction in cellulose content across all organs, indicating that the heightened activity of cellulases and the increased sugar accumulation in the stems may contribute to this phenomenon. This effect is particularly evident in *S. officinarum*, where cellulose exhibits a downward decreasing pattern in the stems, while sugar accumulation peaks at the stem base.

### 2.5. Genetic Transformation of SoCesA10 and SoCesA11 Genes in Rice

We obtained 28 and 26 over-expressing positive plants of *SoCesA10* and *SoCesA11* rice, respectively (Appendix A). The results of the over-expression level detection (Figure 4A,B) showed that the expression levels of the transgenic plants of *SoCesA10* and *SoCesA11* were much higher than those of the wild-type plants. There were no phenotypic differences compared to wild-type plants (Figure 4A,B) throughout the entire growth cycle of rice. Still, the cellulose content was significantly higher than that of wild-type plants. It showed that the levels of *SoCesA10* and *SoCesA11* expression are positively correlated with cellulose content in the transgenic rice lines expressing each gene (Figure 4C).

## 3. Discussion

### 3.1. Comparative Genomics Reveals Organ-Specific Expression and Interspecies Differential Expression of CesA Genes in Sugarcane

Comparative genomics is often used in crop improvement and breeding [21]. By comparing the genomes of different species, we can better understand the patterns and processes behind plant genome evolution and reveal functional regions of the genome [22]. The cellulose synthase (*CesA*) gene family has been identified and studied for its gene function in species such as barley [23], sorghum [24], rice [19], and maize [25]. It has been found that different *CesA* family members exhibit distinct gene expression patterns and functions across species. In this study, based on the existing reference genomes of sugarcane varieties AP85-441 [3] and LA-purple, the *CesA* gene family members were identified at the whole-genome level, and interspecies differences in the structure of the identified *CesA* genes were analyzed. *CesA* genes were found to exhibit organ-specific expression and interspecies differential expression. *CesA1*, *CesA7*, *CesA8*, and *CesA9* exhibit the same expression pattern, which may be related to the deposition of primary cell walls, with the highest expression levels in the upper sections of the stem. The expression of *CesA1* and *CesA7* is higher in the *S. officinarum* compared to the *S. spontaneum*, while the expression of *CesA8* and *CesA9* is higher in the *S. spontaneum* compared to the *S. officinarum*. This results in lower expression of *CesA8* and *CesA9* in the *S. officinarum* compared to the *S. spontaneum*. *CesA10*, *CesA11*, and *CesA12* have consistent expression trends and peak expression at the base of stem internodes, with much higher expression levels in the *S. spontaneum* compared to the *S. officinarum*. The extremely low expression level of *CesA2* may be attributed to its exclusive expression during specific developmental stages and under certain environmental conditions, rendering it undetectable in transcriptome data. The RT-qPCR validation results and the determination of cellulose content across different organs and developmental stages are consistent with these expression trends, indicating that *CesA* genes are involved in regulating cellulose synthesis in sugarcane.

### 3.2. The Regulatory Role of the CesA Gene in Plant Cellulose Synthesis and Growth

Sugarcane is primarily cultivated for its stalks, where sugar and fiber are the main accumulations, making the analysis of fiber components crucial. Different sections of the sugarcane stalk exhibit distinct physiological and metabolic characteristics, such as the progressively higher sucrose content from top to bottom [26]. This is a result of the conversion of excess photosynthetic products into sugar [27]. The metabolism of fiber components is closely intertwined with cell wall synthesis, and as sugarcane develops, the composition and content of fiber components in the cell wall undergo change [28]. The maturity of sugarcane stalk cells is influenced by their position and growth duration, which accounts for the significant variation in fiber components within different organs at various developmental stages.

In plants, defects in cellulose synthesis generally lead to a decrease in cellulose content, resulting in a phenotype of brittle or weak stems [29]. Since the first identification of *CesA* genes, CesA mutants have been reported in various species [12,30,31], almost all of which exhibit varying degrees of defective phenotypes in plant growth and development. This indicates that *CesA* genes regulate the growth and development of plants and are indispensable during the process [12,32]. *CesA* genes regulate cellulose synthesis at the transcriptional level and play an important role in plant cell wall synthesis. We observed that the cellulose content in rice leaves increased with the expression levels of *SoCesA10* and *SoCesA11* in both transgenic rice lines. In rice, recessive mutations in *OsCesA4* [33,34], *OsCesA7* [35], and *OsCesA9* [36] lead to decreased mechanical strength and cellulose proportion, as well as increased hemicellulose content in the stem. This suggests that the functions of *OsCesA4*, *OsCesA7*, and *OsCesA9* are not redundant, and their functional impairment leads to a decrease in cellulose content. Studies have shown a correlation between cellulose content and plant stem mechanical strength [37]. This also implies that these three genes, to some extent, positively regulate stem mechanical strength. In sugarcane, there have also been reports of a significant positive correlation between cellulose content and stem puncture force [38]. However, current research on the CesA gene family in sugarcane is relatively scarce, and existing reports mainly focus on areas such as molecular marker development [39] and gene function prediction (e.g., *CesA7*) [40]. *SoCesA10* and *SoCesA11* are directly homologous genes of *OsCesA7* and *OsCesA4*. The elevated expression of *SoCesA10* and *SoCesA11* in mature stem tissues suggests that these genes may be specifically involved in secondary cell wall thickening, consistent with the roles of *OsCesA4*, *OsCesA7*, and *OsCesA9* in rice. This indicates a conserved mechanism among grasses for cellulose deposition in structural tissues. Moreover, the interspecies differences between *S. spontaneum* and *S. officinarum* highlight potential targets for improving mechanical strength and sugar yield in cultivated sugarcane. This will lay the foundation for breeding sugarcane with high biomass and lodging resistance.

## 4. Materials and Methods

### 4.1. Plant Materials, RNA Extraction, and Sequencing

Two ancestral *Saccharum* species, *S. spontaneum* AP85-441 (Ss, 2n = 8x = 64) and *S. officinarum* LA-Purple (So, 2n = 8x = 80), were used to investigate gene expression patterns. Planted on the campus of Fujian Agriculture and Forestry University (26°04′53.99″ N, 119°13′51.92″ E ). We collected samples during periods of consistent plant growth, with AP85-441 showing a faster growth rate compared to LA-Purple.

The RNA sequencing samples from two accessions (AP85-441 and LA-Purple) were collected at three distinct time points: seeding stage (35 days), premature stage (9 months), and mature stage (12 months). At the seedling stage, leaf and stem tissues were collected from both accessions. At the premature stage and mature stage, five tissue types were sampled: leaves, leaf rolls, upper stems (3rd internode for both accessions), middle stems (6th internode for AP85-441, 9th internode for LA-Purple), and basal stems (9th internode for AP85-441, 15th internode for LA-Purple). Internode selection followed the Moore sampling method to ensure physiological comparability across developmental stages. Each sample had three biological replicates, which were immediately frozen in liquid nitrogen and stored at −80 °C for subsequent RNA extraction. The sampling protocol was implemented in accordance with established procedures described in previous studies [41].

Total RNA was isolated from various sugarcane tissue samples using the Trizol Reagent kit (Invitrogen, Waltham, MA, USA), following the manufacturer’s protocol. The protocol was performed as follows: Tissue samples (50 mg) were pulverized under liquid nitrogen and homogenized in 1 mL of TRIzol reagent on ice, followed by a 5 min incubation. Subsequently, 0.2 mL of chloroform was added, and the mixture was vigorously vortexed for 15 s, incubated for 5 min, and centrifuged at 12,000× *g* for 15 min at 4 °C. After phase separation, the upper aqueous layer was carefully collected. An equal volume of isopropanol was added to the aqueous phase, mixed by inversion, and incubated at room temperature for 5 min, followed by centrifugation at 12,000× *g* for 10 min at 4 °C. The supernatant was discarded, and the resulting RNA pellet was washed with 1 mL of 75% ethanol, centrifuged at 7500× *g* for 5 min at 4 °C. After complete removal of the supernatant, the pellet was briefly centrifuged again to remove residual ethanol and air-dried for 5 min at room temperature. Finally, the purified RNA was dissolved in 20 μL of nuclease-free water and stored at −80 °C for subsequent analysis. The cDNA libraries for each sample were constructed using Illumina® TruSeq™ RNA Sample Preparation Kit (RS-122–2001 (2), Illumina (Illumina, San Diego, CA, USA)). The libraries were sequenced using an Illumina HiSeq2500 instrument with paired-end 100 bp reads. Fastp was utilized for quality control, with parameters set to -qualified_quality_phred 10 and -unqualified_percent_limit 8. Furthermore, the gene expression abundance was quantified by transcripts per kilobase of exon model per million mapped reads (TPM) values. RT-qPCR primers were designed based on the coding sequence (CDS) of *CesA10* and *CesA11* genes from *S. officinarum*. GAPDH (glyceraldehyde-3-phosphate dehydrogenase) and 25S rRNA (25S ribosomal RNA) were used as dual reference genes. The primer sequences used are shown in Table 2. All transcriptome data is accessible at: https://sugarcane.gxu.edu.cn/download/AP85-441/AP85-441.tissues.tar.gz (accessed on 27 October 2025).

### 4.2. Identification and Physicochemical Analysis of Cellulose Synthase Gene Family

Using the Sorghum *CesAs* gene sequence as a reference, a chromosome-level genome sequence BLAST search was conducted [2]. The matches from the LA-purple chromosome whole genome (Sugarcane Genome Database (gxu.edu.cn (accessed on 27 October 2025))) BLAST search with similarity scores > 80.0 and coverage > 90.0 were selected as candidate genes for *S. officinarum CesA*. To ensure reliable results, chromosome location detection was performed for all candidate *CesA* genes to remove redundant gene sequences located on the same chromosome. Additionally, the online tool ExPasy (http://www.expasy.org/ (accessed on 27 October 2025)) was used for the calculation of physicochemical properties of CesA proteins, including the number of amino acids, molecular weight, and isoelectric point. Subcellular localization was predicted using online software analysis tool ProtComp9.0 (http://www.softberry.com/berry.phtml (accessed on 27 October 2025)). The number of alleles was determined based on distinct gene copies identified through BLAST search and chromosomal location analysis within the *S. officinarum* genome.

### 4.3. Gene Structure, Protein Conserved Motifs, and Phylogenetic Analysis

Sequences of *CesA* family members from *Hordeum vulgare*, *Zea mays*, *Vitis vinifera*, *Sorghum bicolor*, *Arabidopsis thaliana*, and *Oryza sativa* were obtained from the NCBI and Phytozome v13 databases. For detailed information, please refer to Appendix A. Multiple sequence alignment of CesA proteins was performed using MAFFT. The phylogenetic tree based on the alignments was inferred using the neighbor-joining method implemented in MEGA X [42]. Bootstrap analysis was performed with 1000 replicates. Additional maximum-likelihood (ML) phylogenetic trees were constructed using iq-tree with the following parameters: iqtree -nt AUTO -bb 1000, analyze gene structure using the online tool GSDS (http://gsds.cbi.pku.edu.cn/ (accessed on 27 October 2025)), and identify conserved motifs of CesA proteins using the online tool MEME.

### 4.4. Measurement of Cellulose Content in S. spontaneum and S. officinarum

The cellulose content in different organs at different developmental stages (See Section 2.1 for details) of the two original species of sugarcane was determined using the cellulose (CLL) content assay kit (COMIN: CLL-2-Y) from a commercial company (Suzhou Comin Biotechnology Co., Ltd., Suzhou, China). The rice samples were taken from the second fully expanded leaf at the maturity stage, and the measurements were conducted using the same method as described above. The specific steps were followed according to the instruction manual (http://www.cominbio.com/uploads/soft/220531/1-220531095000.pdf (accessed on 27 October 2025)). The brand and model of the spectrophotometer is BioTek Epoch2 (Agilent, Santa Clara, CA, USA).

### 4.5. Subcellular Localization and Genetic Transformation

The full-length cDNA of *SoCesA10* and *SoCesA11* was obtained from the cDNA of LA-Purple using specific primers (SoCesA10-L: ATGGACACCGGCTCGGT, SoCesA10-R: GCACTCGACTCCGC, SoCesA11-L: ATGGAGTCGGCGGCGGC, SoCesA11-R: GACTGTGTTGCAGTTGTTGGT) that were integrated into the *Sal I* and *Kpn I* linearized pSuper1300-GFP vector (Appendix A) using In-Fusion^®^ HD Cloning Kit (639648, Takara, Beijing, China), following the manufacturer’s instructions (https://www.takarabio.com/assets/a/112472 (accessed on 27 October 2025)).

An overnight *Agrobacterium* (strain GV3101) culture was harvested and resuspended in infiltration buffer consisting of 10 mM MgCl_2_, 10 mM MES (pH 5.4), and 100 μM acetosyringone (AS), to an optical density of OD_600_ = 0.5. The Agrobacterium strain used was GV3101 (containing the helper plasmid pSoup, Weidi Biotechnology Co., Ltd., Shanghai, China). The suspension was incubated at 25 °C for 2 h to activate virulence genes. Subsequently, the Agrobacterium suspension was infiltrated into the abaxial side of 4-week-old *N. benthamiana* leaves using a 1 mL needleless syringe. *N. benthamiana* plants were grown under controlled conditions (25 °C, 16 h light/8 h dark photoperiod) in an artificial climate chamber. After infiltration, the plants were incubated for 2 days under the same conditions to allow expression of the fluorescent fusion protein. Confocal images were captured with a Leica TCS SP8X DLS (OLYMPUS, Tokyo, Japan). In confocal imaging, a 60× objective lens with a numerical aperture of 1.4 was utilized, an excitation source of 488 nm and a filtering of 505 nm, and a PMT detector. The exposure time was set to 500 ms.

Genetic transformation was performed using wild-type *Oryza sativa* ssp. *Japonica* (Nipponbare) as the background material. Xcm I-F (5′-GATGATAAGCCAATACTT-3′) adapter was added to the upstream primer, and XcmI-R (5′-ATTCGGATCCCCAATACCTA-3′) adapter was added to the downstream primer, synthesized by Fuzhou Shangya Biotechnology Co., Ltd. The amplified products were ligated to pCXUN-FLAG vector (Appendix A), transformed into competent *Escherichia coli* cells, and positive clones were selected, and the positive plasmids were recovered. Double digestion was performed using Xcm I (NEB#: R0533S) restriction enzyme. Rice genetic transformation was performed by Wuhan Boyuan Biotechnology Co., Ltd. (Wuhan, China). The detailed operating procedures are as follows: (1) Rice grains with no mildew spots and normal bud mouths were selected, disinfected with 75% alcohol for 1 min, and rinsed with sterile water once for 1 min; then disinfected with 15% sodium hypochlorite for 20 min and rinsed with sterile water three times for 1 min each time. The disinfected rice grains were inoculated on the induction medium and cultured under light at 26 °C for 20 days. (2) *Agrobacterium* was picked into the infection solution to prepare a resuspension with OD_600_ = 0.2. Calli were placed in an Erlenmeyer flask, the *Agrobacterium* resuspension was added, and infection was carried out for 10–15 min, after which the bacterial solution was discarded. The calli were inoculated on the co-cultivation medium and co-cultivated at 20 °C for 48–72 h. (3) Calli from step (2) were inoculated on the screening medium and cultured in the dark at 26 °C for 20–30 days; positive calli were then inoculated on the secondary screening medium. During the process of picking calli, monoclonal calli were carefully selected and cultured in the dark at 26 °C for 7–10 days. (4) Positive calli were inoculated on the differentiation medium and cultured under light at 25–27 °C for 15–20 days. After buds of 2–5 cm were differentiated, they were inoculated on the rooting medium and cultured under light at 30 °C for 7–10 days. The planting conditions for rice included a room temperature of 25 °C and a light/dark cycle of 16 h of light and 8 h of darkness. The seeds were placed in a growth chamber under light, and a nutrient solution was used to culture the plants for half a month. Positive seedlings of the T_0_ generation were identified by PCR and RT-qPCR.

## 5. Conclusions

Ten *CesA* family members identified in the *S. officinarum* were grouped into six clades, among which *SoCesA10*, *SoCesA11*, and *SoCesA12* clustered with secondary cell wall synthesis-related genes in rice and Arabidopsis, while the other genes, except for *CesA2*, clustered with primary cell wall synthesis-related genes. This suggests that the six genes related to the primary cell wall may have functional redundancy, while the three genes related to secondary cell wall synthesis may play an indispensable role in cellulose synthesis. Combining the results of RT-qPCR expression patterns at different developmental stages and cellulose content determination, it can be inferred that *CesA* genes positively regulate cellulose synthesis.

## Figures and Tables

**Figure 1 ijms-26-11046-f001:**
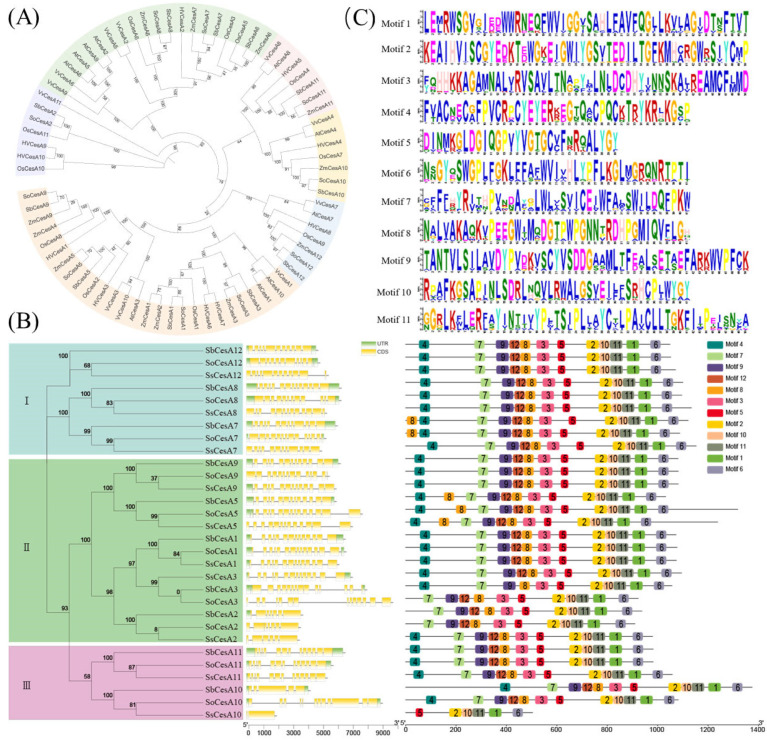
Phylogenetic tree of CesA. (**A**) Phylogenetic tree of CesA protein family from *S. officinarum*, *Sorghum bicolor*, *Zea mays*, *Hordeum vulgare*, *Vitis vinifera*, *Arabidopsis thaliana*, and *Oryza sativa* (Gene ID in Appendix A). The number represents the confidence level of the branch and displays the phylogenetic relationship between different samples. The branch lengths represent evolutionary distance; the longer the branch, the greater the difference or evolutionary time between the two samples. (**B**) Phylogenetic, gene structure analysis, and conserved protein motif analysis of *CesA* genes in sorghum and sugarcane (*S. officinarum* and *S. spontaneum*). (**C**) Sequence analysis of each conserved motif of *CesA* protein in *S. officinarum*, *S. spontaneum,* and *Sorghum bicolor*.

**Figure 2 ijms-26-11046-f002:**
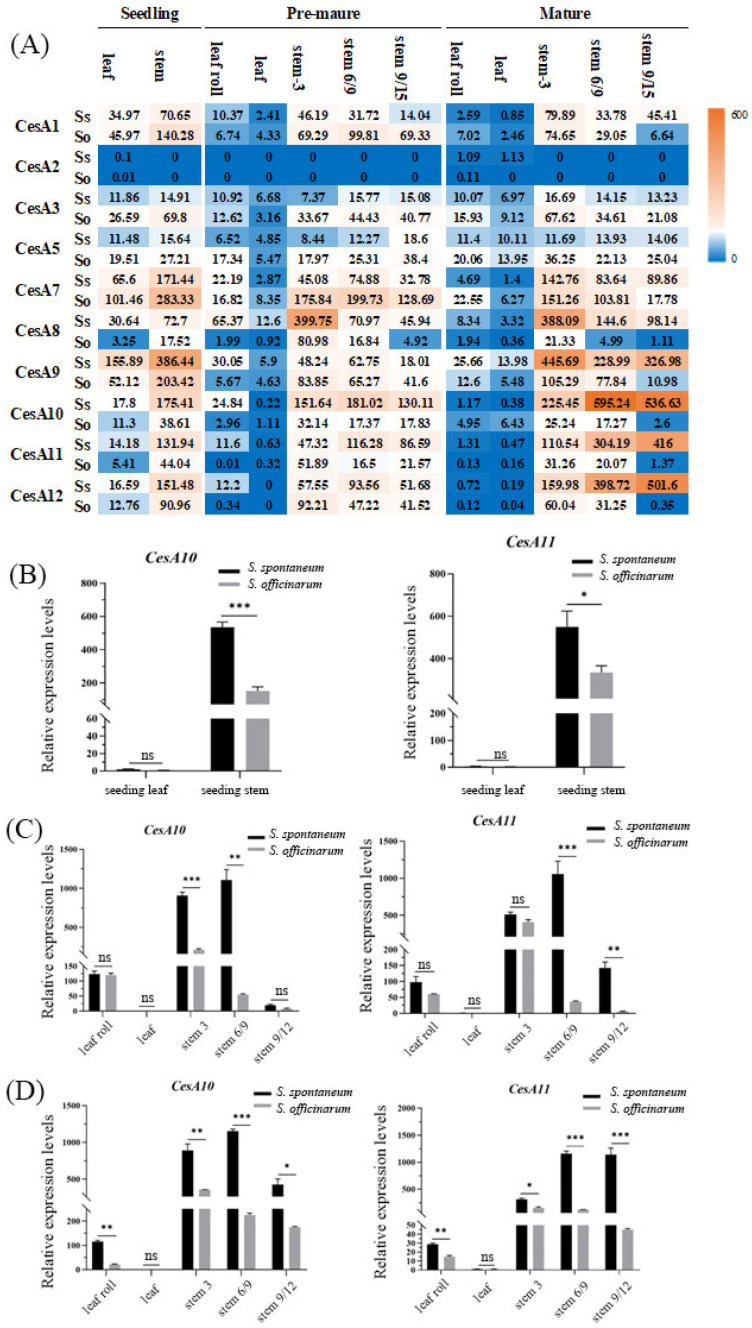
Transcriptome and RT-qPCR validation of *CesA* genes expression across genotypes and developmental stages. (**A**) heat map showing the expression levels of 10 CesA genes in two sugarcane species, AP85-441 (Ss) and LA-Purple (So), across three developmental stages. The values in the heat map represent TPM values. (**B**–**D**) Represent the RT-qPCR validation results of expression patterns of *CesA10* and *CesA11* genes in different tissues during the seedling stage, pre-mature stage, and mature stage of the *S. spontaneum* and *S. officinarum*. Error bars, SD (*n* = 3). * indicates *p*-value < 0.05, ** indicates *p*-value < 0.01 and *** indicates *p*-value < 0.001. ns indicates no significant difference.

**Figure 3 ijms-26-11046-f003:**
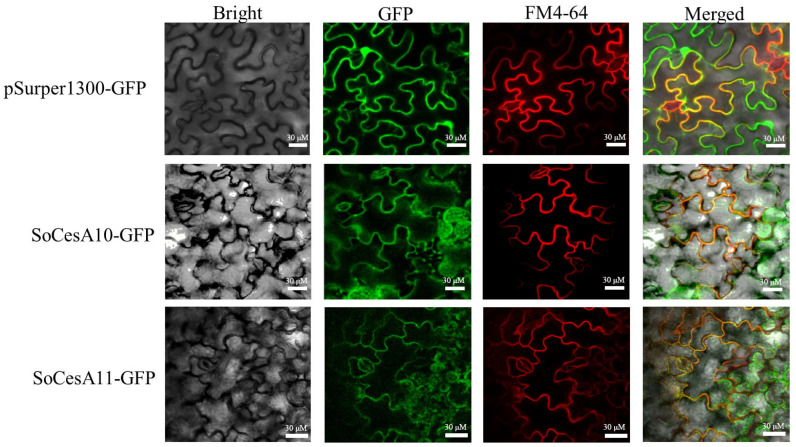
The subcellular localization of SoCesA10 and SoCesA11 in tobacco epidermal cells and the white scale in each picture is 20 μM GFP signals (green) colocalized with the plasma membrane marker FM4-64 (red), indicating that SoCesA10-GFP and SoCesA11-GFP localize to the plasma membrane.

**Figure 4 ijms-26-11046-f004:**
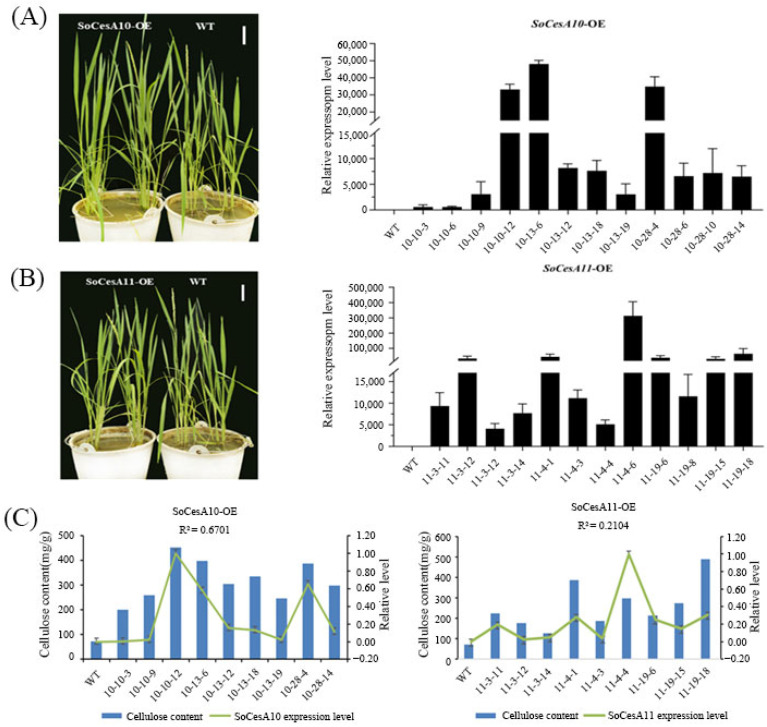
Analysis of leaf cellulose content and expression level in Rice overexpressed Plants. (**A**,**B**) The phenotypes and expression levels of *SoCesA10* and *SoCesA11* in overexpression positive plants, respectively. All plants were at the mature stage with no significant difference in plant height. (**C**) Leaf cellulose content of overexpression rice plants. Error bars, SD (*n* = 3). R^2^, the coefficient of determination, indicates the correlation between Cellulose and the expression level.

**Table 1 ijms-26-11046-t001:** The identification and protein physico-chemical analysis of *S. officinarum CesA* gene.

Gene Name	Gene ID	Number of Alleles	Number of Amino Acids	Theoretical pI	Molecular Weight	Aliphatic Index	Grand Average of Hydropathicity	Instability Index	Subcellular Localization
*SoCesA1*	Soffic.09G0009150	5	1076	6.6	121,187.11	84.89	−0.195	39.89	Plasma membrane
*SoCesA2*	Soffic.10G0017310	6	937	6.7	102,705.03	87.39	−0.126	52.04	Plasma membrane
*SoCesA3*	Soffic.03G0004580	9	1094	6.12	123,061.17	88.92	−0.154	36.93	Plasma membrane
*SoCesA5*	Soffic.01G0017790	14	1317	8.47	147,389.74	83.35	−0.261	41.1	Plasma membrane
*SoCesA7*	Soffic.02G0008760	8	1087	6.52	122,887.03	81.15	−0.245	39.78	Plasma membrane
*SoCesA8*	Soffic.02G0007110	3	1095	7.42	122,601.02	82.55	−0.186	44.61	Plasma membrane
*SoCesA9*	Soffic.02G0032990	8	1081	8.04	120,986.5	81.34	−0.231	37.39	Plasma membrane
*SoCesA10*	Soffic.01G0033860	3	1374	8.69	154,548.22	76.8	−0.316	39.45	Plasma membrane
*SoCesA11*	Soffic.02G0033140	6	981	6.22	110,255.67	83.9	−0.115	39.84	Plasma membrane
*SoCesA12*	Soffic.02G0014650	6	1051	6.36	118,141.03	84.67	−0.158	36.17	Plasma membrane

**Table 2 ijms-26-11046-t002:** Primers sequences used for RT-qPCR.

Primer Name	Left Primer	Right Primer
*GAPDH*	CACGGCCACTGGAAGCA	TCCTCAGGGTTCCTGATGCC
*25S rRNA*	GCAGCCAAGCGTTCATAGC	CCTATTGGTGGGTGAACAATCC
*SoCesA10*	GTCATCAGCTGCGGATACGA	GTGCAGTACACCGACTTCCA
*SoCesA11*	CATGGCCTGGGAACAATCCT	TGTCAGAACAGCGGACACTC

## Data Availability

The original contributions presented in this study are included in the article. Further inquiries can be directed to the corresponding author.

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
