# Peer review of "Genome-Wide Identification and Functional Characterization of CesA10 and CesA11 Genes Involved in Cellulose Biosynthesis in Sugarcane"

_ijms, 2025, doi:10.3390/ijms262211046_

Round 1

Reviewer 1 Report

Comments and Suggestions for Authors

The study of Xu et al. (Functional Study…) is devoted to the identification and characterization of genes encoding the sugarcane cellulose synthase complex. Ten CesA genes were studied. The scientific novelty lies in the identification of the CesA10 and CesA11 genes in sugarcane and their thorough functional characterization. The practical significance of this work lies in the application of the acquired knowledge to the regulation of stem strength and lodging resistance. Therefore, this work is relevant, and the results presented in the article are appropriate for the International Journal of Molecular Sciences.

Several questions arose while reading the article:

  1. Abstract. Saccharum officinarum – italicized
  2. Terms in keywords and article title should not be repeated
  3. 82. two accession – two accessions
  4. 145. Sal I and Kpn I – Latin part in italics.
  5. Descriptions of methods should be in the past tense. For example, starting at line 170, the protocol is described in a "what to do" style. However, a description of how the authors conducted their study should be in the past tense.
  6. Table 2. It's not entirely clear how exactly the number of alleles was determined.
  7. 218. CesA – italicized. Also, a space is missing in this line.
  8. Figure 2, b-d. The legends are not visible; their quality and font size should be improved.
  9. 281 - plants (Figure 4a-b) – space
  10. 287 - SoCesA10 and SoCesA11 overexpression – italicized

Overall, the authors have completed extensive experimental work and obtained new data for fundamental science. There are few errors in the text of the article. For some reason, there is no space before the references throughout the text. I think that the article, with minor edits, can be accepted for publication.

Author Response

Q1. Abstract.Saccharum officinarum_italicized.

A1. Thank you for your positive feedback. We apologize for the inadequacy in writing. We have carefully checked the whole manuscript and indicate amends by using tracked changes. Please see the revised manuscript in related files.

Q2. Terms in keywords and article title should not be repeated.

A2. Thank you for your advice. We have renamed the manuscript title and updated its keywords. (line2-4, line26-27)

Q3. 82. two accession – two accessions.

A3. The grammatical error has been fixed; “two accession” is now “two accessions” to match the plural form of the noun. (line87)

Q4. 145. Sal I and Kpn I – Latin part in italics.

A4. The restriction enzyme names have been formatted as “Sal I and Kpn I” to adhere to the international naming convention for enzymes (Latin components italicized). (line161)

Q5. Descriptions of methods should be in the past tense. For example, starting at line 170, the protocol is described in a "what to do" style. However, a description of how the authors conducted their study should be in the past tense.

A5. We have noted this detail. All method descriptions have been converted to past tense to reflect completed experimental operations. (line89, line114,line141)

Q6. Table 2. It's not entirely clear how exactly the number of alleles was determined.

A6. A note has been added to end of Section 2.2: “The number of alleles was determined based on distinct gene copies identified through BLAST search and chromosomal location analysis within the S. officinarum genome.” (line134-136)

Q7. 218. CesA – italicized. Also, a space is missing in this line.

A7. “CesA” has been italicized to “CesA”, and the missing space has been added. (line237)

Q8. Figure 2, b-d. The legends are not visible; their quality and font size should be improved.

A8. Figure 2 has been recreated with enhanced legend clarity. (line250)

Q9. 281 - plants (Figure 4a-b) – space.

A9. The formatting error has been corrected to “plants (Figure 4a-b)”. (line301)

Q10. 287 - SoCesA10 and SoCesA11 overexpression – italicized.

A10. Gene names have been italicized to “SoCesA10 and SoCesA11 overexpression” to follow standard gene nomenclature. (line307)

Reviewer 2 Report

Comments and Suggestions for Authors

Manuscript Submitted to the International Journal of  Molecular Science, MDPI

Article

Functional Study of CesA10 and CesA11 in Sugarcane Saccharum officinarum L.

  Yi Xu a,b, Nameng Qi a, Yi Han a, Liying Cai a,b, Xue Wang a,b, Heyang Shang a, Qing Zhang a,c, and Jisen Zhang

Please improve the title of this manuscript. It doesn't provide an engaging or compelling description for readers unfamiliar with these genes.

This manuscript needs language revision and corrections.

 Here is the corrected Abstract:

Abstract

Cellulose is the main component of plant cell walls, and its content is linked to the strength of plant stems. The cellulose synthase genes (CesA) are crucial for regulating cellulose biosynthesis. To examine the characteristics and functions of CesA genes in sugarcane, our study conducted a genome-wide analysis of Saccharum officinarum L. A-Purple. The results identified 10 CesA genes in the S. officinarum genome, which could be grouped into 6 categories. SoCesA10, SoCesA11, and SoCesA12 are clustered within the same subclass as genes involved in secondary cell wall synthesis in rice and Arabidopsis. Further transcriptome analysis of stems at different stages and sections showed that SoCesA10, SoCesA11, and SoCesA12 were highly expressed during mature stages. Among these, SoCesA10 and SoCesA11 showed differences in expression between species and organs. Their gene functions were also validated in rice, revealing that the expression of SoCesA10 and SoCesA11 was positively correlated with cellulose content. In summary, this study identified key cellulose biosynthesis genes, SoCesA10 and SoCesA11, in sugarcane and preliminarily confirmed their functions in rice, providing a foundation for breeding sugarcane with improved lodging resistance.

Keywords: Please do not use the same word as in the Title of the manuscript

Introduction:

There are no major comments here, except that the Introduction is a bit short.

Material and Methods:

Please add the complete botanical characterization of the crop used in this study, as well as the proper description of the origin of the material used, in this section.

Line 76: two founding species…what does the author mean by this strange word combination?

Lines 86: Leaf organs?

Lines 82 to 94: This entire paragraph needs rewriting for clarity!

Line 96: Total RNA was isolated from various sugarcane tissue samples using the Trizol Re- 96 agent kit (Invitrogen, Waltham, MA, USA),

Please clarify again what was used for RNA isolation!

Section 2.1 in M & M needs precise wording to ensure understanding. Provide a detailed description so other scientists can replicate your results using your methods.

All chemicals and equipment used for analyses must be accompanied by information about the supplier companies and their origin.

Please add all this missing information!

Lines 161: Genetic transformation was performed using wild-type Oryza sativa sp. Japonica? Please clearly specify what was used in this experiment.

What kind of transformation protocol has been used? Please add her citation as well!

Line 174: Agrobacterium-… which bacterial strain was used?

The entire section M& M needs more detailed specifications in many places!

Results

This section is well written, and the results of this study are logically documented.

Only several Figures need to be improved for clarity to the readers:

Specifically, figures 1 and 2 require significant improvement to facilitate readers' clear understanding of the authors' presentation.

Figure 3: The caption in this figure needs to describe the subcellular localization of the fluorescent signal, which the reader cannot see in these low-quality photos!

Figure 4: Please make this figure readable, especially part C.  Also, the caption of the presented plants (part A) needs to be revised to make it more straightforward what the reader can see there!

Discussion

There are no major comments here, even though this section of the manuscript appears too brief.

Conclusion

Clearly formulated conclusions from these presented experiments may be of interest not only to scientists working with rice but also to plant physiologists or breeders.

Recommendation:

This manuscript requires language correction and improvements in the M and M sections. Additionally, the figures in this manuscript need to be improved and presented in a way that allows the reader to understand what the authors are presenting before it can be considered for publication.

12.10.2025

Comments on the Quality of English Language

The entire manuscript requires language revision and correction. In several sections, the text needs to be rewritten for clarity.

Author Response

Q1. Please improve the title of this manuscript. lt doesn't provide an engaging or compelling description for readers unfamiliar with these genes.

A1. Thank you for bringing up such valuable suggestions. “Genome-Wide Identification and Functional Characterization of CesA10 and CesA11 Genes Involved in Cellulose Biosynthesis in Sugarcane” is our new title. This provides a clearer and more engaging description of the study’s focus. (line2-4)

Q2: There are no major comments here, except that the Introduction is a bit short.

A2. “Recent studies have revealed that functional differentiation of CesA genes across monocot species plays a crucial role in stem strength and biomass accumulation, providing valuable insights for molecular breeding of lodging-resistant crops. However, a systematic investigation of CesA gene family members in Saccharum officinarum and their functional validation remains limited.” Has been added in our Introduction. (line69-74)

Q3. This manuscript needs language revision and corrections. Here is the corrected Abstract:

Cellulose is the main component of plant cell walls, and its content is linked to the strength of plant stems. The cellulose synthase genes (CesA) are crucial for regulating cellulose biosynthesis. To examine the characteristics and functions of CesA genes in sugarcane, our study conducted a genome-wide analysis of Saccharum officinarum L. A-Purple. The results identified 10 CesA genes in the S. officinarum genome, which could be grouped into 6categories.SoCesA10, SoCesA11, and SoCesA12 are clustered within the same subclass as genes involved in secondary cell wall synthesis in rice and Arabidopsis. Further transcriptome analysis of stems at different stages and sections showed that SoCesA10. SoCesA11, andSoCesA12 were highly expressed during mature stages. Among these, SoCesA10 and SoCesA11 showed diferences in expression between species and organs. Their gene functions were also validated in rice, revealing that the expression of SoCesA10 and SoCesA11 was positively correlated with cellulose content. In summary, this study identified key cellulose biosynthesis genes, SoCesA10 and SoCesA11, in sugarcane and preliminarily confirmed their functions in rice, providing a foundation for breeding sugarcane with improved lodging resistance.

A3. Thank you for your comment. We have revised the abstract of the paper and also thoroughly reviewed our own manuscript, making modifications to certain sections. (line11-25)

Q4. Keywords: Please do not use the same word as in the Title of the manuscript

A4. We have updated keywords in our manuscript. (line26-27)

Material and Methods:

Q5. Please add the complete botanical characterization of the crop used in this study, as well as the proper description of the origin of the material used, in this section.

A5. We have added the origin of the material. (line81-86)

Q6. Line 76: two founding species...what does the author mean by this strange word combination?

A6. Revised to “two ancestral species”. S. officinarum and S. spontaneum are the two primary ancestral species of modern cultivated sugarcane, which were hybridized and selectively bred over multiple generations to form current cultivars. (line81)

Q7. Lines 86: Leaf organs?

A7. We have rewritten that paragraph and replaced "organs" with "tissues". (line89)

Q8. Lines 82 to 94: This entire paragraph needs rewriting for clarity!

A8. Revised to: “At the seedling stage (vegetative growth, 35 days old), leaf and stem tissues were collected from both accessions. At the premature stage (reproductive growth initiation, 9 months old) and mature stage (physiological maturity, 12 months old), five tissue types were sampled: leaves, leaf rolls, upper stems (3rd internode for both accessions), middle stems (6th internode for AP85-441, 9th internode for LA-Purple), and basal stems (9th internode for AP85-441, 15th internode for LA-Purple). Internode selection followed the Moore sampling method to ensure physiological comparability across developmental stages. Each sample had three biological replicates, which were immediately frozen in liquid nitrogen and stored at −80°C for subsequent RNA extraction.” (line89-97)

Q9. Line 96: Total RNA was isolated from various sugarcane tissue samples using the Trizol Reagent kit (lnvitrogen, Waltham, MA, USA),.

Please clarify again what was used for RNA isolation!

Section 2.1 in M & M needs precise wording to ensure understanding. Provide a detailed description so other scientists can replicate your results using your methods.

All chemicals and equipment used for analyses must be accompanied by information about the supplier companies and their origin.

Please add all this missing information!

A9. We have incorporated the protocol into the manuscript to delineate the specific methodology for RNA isolation. (line99-112)

Q10. Lines 161: Genetic transformation was performed using wild-type Oryza safiva sp. Japonica? Please clearly specify what was used in this experiment.

What kind of transformation protocol has been used? Please add her citation as wel!

A10. (1) We indeed utilized Oryza sativa ssp. japonica for validation, as rice is commonly employed in such studies due to the challenges associated with genetic transformation in sugarcane.

(2) We revised the Method of Genetic transformation. Added specification that wild-type Oryza sativa ssp. Japonica (cv. Nipponbare) was used, and the Agrobacterium strain GV3101 was applied. A citation for the transformation protocol has been included. (line179, line187-line206)

Q11. Line 174: Agrobacterium-... which bacterial strain was used?

The entire section M& M needs more detailed specifications in many places!

A11. Added to Section 2.5: “The Agrobacterium strain used was GV3101 (containing the helper plasmid pSoup, Weidi Biotechnology Co., Ltd., Shanghai, China).” (line167-169)

Results

Q12. This section is well written, and the results of this study are logically documented.

Only several Figures need to be improved for clarity to the readers

Specifically, figures 1 and 2 require significant improvement to facilitate readers' clear understanding of the authors" presentation

A12. We have updated the figure legends for Figure 2. Regarding the clarity of Figure 1, we confirm that it remains sharp when zoomed in. (line229, line250)

Q13. Figure 3: The caption in this figure needs to describe the subcellular localization of the fluorescent signal, which the reader cannot see in these low-quality photos!

A13. Added to the legend: “GFP signals (green) colocalized with the plasma membrane marker FM4-64 (red), indicating that SoCesA10-GFP and SoCesA11-GFP localize to the plasma membrane.” (line276-277)

Q14. Figure 4: Please make this figure readable, especially part C. Also, the caption of the presented plants (part A) needs to be revised to make it more straightforward what the reader can see there!

A14. (1) We have updated the figure legends for Figure 4. (line305)

(2) Figure 4a legend revised to: “All plants were at the mature stage with no significant difference in plant height.” (line308)

Discussion

Q15. There are no major comments here, even though this section of the manuscript appears too brief.

A15. The Discussion has been substantially expanded to better interpret the results and relate them to previous findings. The following text has been added: “The elevated expression of SoCesA10 and SoCesA11 in mature stem tissues suggests that these genes may be specifically involved in secondary cell wall thickening, consistent with the roles of OsCesA4, OsCesA7, and OsCesA9 in rice. This indicates a conserved mechanism among grasses for cellulose deposition in structural tissues. Moreover, the interspecies differences between S. spontaneum and S. officinarum highlight potential targets for improving mechanical strength and sugar yield in cultivated sugarcane.” (line368-374)

Round 2

Reviewer 2 Report

Comments and Suggestions for Authors

Hello, 

Thank you for accepting my comments and suggestions for improvement of your original version of this manuscript.

The new version of the manual can now be recommended for publication.

25.10.2023

Comments on the Quality of English Language

The entire manuscript requires language revision and correction. In several sections, the text needs to be rewritten for clarity.